# An integrated analysis of the structural changes and gene expression of spleen in human visceral leishmaniasis with and without HIV coinfection

Jonathan L. M. Fontes [1,2], Ricardo Khouri[1], Luis Gustavo C. Reinaldo[3], Erina M. A. Hassegawa[1], Antônio José Meneses Filho[4], Caroline V. B. de Melo[1,2], Pablo Ivan P. Ramos[1], Rafael de Deus Moura[5], Carla Pagliari[6], Marta Santos[1], Raimundo José C. Araújo, Jr.[5], Johan Van Weyenbergh[7], Luiz A. R. de Freitas[1], Carlos Henrique N. Costa[4], Washington L. C. dos-Santos [1,2]*

1 Fundação Oswaldo Cruz, Instituto Gonçalo Moniz, Salvador, Bahia, Brazil, 2 Departamento de Patologia e Medicina Legal, Faculdade de Medicina, Universidade Federal da Bahia, Salvador, Bahia, Brazil, 3 Hospital Universitário, Universidade Federal do Piauí, Teresina, Piauí, Brazil, 4 Instituto de Doenças Tropicais Natan Portela, Universidade Federal do Piauí, Teresina, Piauí, Brazil, 5 Departamento de Medicina Especializada, Universidade Federal do Piauí, Teresina, Piauí, Brazil, 6 Faculdade de Medicina, Universidade de São Paulo, São Paulo, São Paulo, Brazil, 7 Laboratory for Clinical and Evolutionary Virology, Ku Leuven, Leuven, Belgium

* washington.santos@fiocruz.br

**Data Availability Statement:** All relevant data are within the manuscript and its Supporting Information files.

## Abstract

The spleen plays a pivotal role in the pathogenesis of visceral leishmaniasis. In severe forms of the disease, the spleen undergoes changes that can compromise its function in surveilling blood-circulating pathogens. In this study, we present an integrated analysis of the structural and gene expression alterations in the spleens of three patients with relapsing visceral leishmaniasis, two of whom were coinfected with HIV. Our findings reveal that the IL6 signaling pathway plays a significant role in the disorganization of the white pulp, while *BCL10* and *ICOSLG* are associated with spleen organization. Patients coinfected with HIV and visceral leishmaniasis exhibited lower splenic CD4+ cell density and reduced expression of genes such as *IL15*. These effects may contribute to a compromised immune response against *L. infantum* in coinfected individuals, further impacting the structural organization of the spleen.

## Author summary

While most individuals recover after contracting visceral leishmaniasis, there are cases where patients experience a relapsing form of the disease. This response pattern to *Leishmania* infection is linked to alterations in spleen structure. Until now, these spleen changes had only been examined in dogs and laboratory animals. In our study, we provide data on the disorganization of spleen structure in three patients diagnosed with VL, two of whom were also co-infected with HIV. The observed patterns of spleen disorganization

**Funding:** This work was supported by the Conselho Nacional de Desenvolvimento Científico e Tecnológico – CNPq (Financial support grant no. 424776/2016-2 to WLCS; Research productivity scholarship to WLCS; Scholarship "Bolsa Jovem Talentos" to RK) and the Coordenação de Aperfeiçoamento de Pessoal de Nível Superior – CAPES (Financial support to WLCS; PhD scholarship to JLFM; PhD scholarship to CVBM). The funders had no role in study design, data collection and analysis, decision to publish, or preparation of the manuscript.

**Competing interests:** The authors have declared that no competing interests exist.

closely resembled those seen in dogs. Our study emphasizes the involvement of IL6, BCL10, and ICOSLG in this process.

## Introduction

Zoonotic visceral leishmaniasis (VL), caused by *Leishmania infantum*, is endemic in Brazil, various parts of the American Continent, and the Mediterranean Basin. The disease manifests through clinical symptoms such as fever, emaciation, hepatosplenomegaly, paleness, pancytopenia, bleeding diathesis, and a weakened immune response to bacterial infections [1,2]. While most patients respond to treatment with pentavalent antimony (Sb$^V$) or amphotericin B, there exists a subset with a poor response to conventional therapy, leading to a relapsing form of the disease [3,4]. Understanding the factors contributing to the progression of VL remains limited, yet HIV co-infection and other immunosuppressive conditions have been linked to a poorer prognosis [5].

The spleen, a significant secondary lymphoid organ, plays a vital role in immune surveillance against blood-circulating pathogens [6,7]. In severe VL cases, the spleen is consistently affected and is thought to underlie key clinical manifestations and disease progression [8,9]. Hypersplenism observed in VL contributes to anemia and may be associated with bleeding tendencies [10–13]. Furthermore, disruption of spleen white pulp (WP) and replacement of cell populations may transform the spleen into a permissive environment for parasite survival and proliferation [9,14–16].

This study investigates the association between clinical presentation, spleen histology, and gene expression in three patients exhibiting relapsing forms of VL unresponsive to conventional therapy with Sb$^V$ or amphotericin B. These patients underwent splenectomy as an adjunctive therapeutic measure. The descriptive approach was expanded by comparing the gene expression profiles with those of health patients from publicly available data. The information presented herein aim to contribute to our understanding of the pathways involved in susceptibility to an inadequate response to *L. infantum* infection.

## Results

### Clinical and laboratory data

Clinical parameters are summarized in Table 1. Infection was confirmed in all patients by the identification of amastigotes in bone marrow aspirates. Each of the three patients had experienced at least one year of visceral leishmaniasis (VL) with a minimum of four relapses, despite undergoing conventional treatment. Patients VL1 and VL3 had been living with HIV infection for nine and seven years, respectively. All patients presented with a palpable spleen below the left costal edge: VL1 (18cm), VL2 (12cm), and VL3 (4cm). Additional clinical information can be found elsewhere [17].

Before splenectomy, all patients experienced anemia and leukopenia. Patients VL1 and VL3 also presented with thrombocytopenia (platelet count less than 150,000/μL). Post-splenectomy, a mild improvement in hemoglobin levels was observed in all patients; however, VL1 and VL3 continued to exhibit signs of anemia. There was an improvement in the blood leukocyte counts for all patients. Additionally, the TCD4+ count increased in patients VL1 and VL3 after splenectomy. Unfortunately, two patients died, as previously described [17]: VL1 due to respiratory distress 244 days post-splenectomy, and VL3 due to severe digestive bleeding 742 days after the splenectomy procedure.

**Table 1. Clinical parameters of VL patients splenectomized due to relapses after conventional treatment failure.**

| | PATIENT | | |
|---|---|---|---|
| **PARAMETERS** | **VL1** | **VL2** | **VL3** |
| **Age** | 33 | 55 | 45 |
| **Sex** | M | M | F |
| **Disease duration** Years living with HIV | 9 | - | 7 |
| Years with VL | 5 | 1 | 3 |
| Relapses | 5 | 5 | 4 |
| **Splenectomy (year)** | 2015 | 2015 | 2015 |
| **Pre-splenectomy parameters:** | | | |
| Bone marrow amastigotes | + | + | + |
| HIV copies | <50 | - | Undetectable |
| Hemoglobin (g/dL) | 6.6 | 8.7 | 9.0 |
| White blood cell count /mm$^3$ | 1170 | 1090 | 1420 |
| CD4 count/mm$^3$ | 33 | - | 104 |
| Platelets/ml ($10^3$) | 115 | 155 | 90 |
| **Post-splenectomy parameters:** Bone marrow amastigotes | ND | ND | + |
| Hemoglobin (g/dL) | 7.8 | 14.5 | 10.1 |
| White blood cell counts | 4840 | 5800 | 5200 |
| CD4 count/mm$^3$ | 135 | - | 436 |
| Platelets/ml ($10^3$) | 194 | 383 | 101 |
| **Additional VL control treatment** | - | - | Intranasal Sb$^V$ |
| **Outcome** | Death at the 244$^{th}$ day post-splenectomy | Clinical cure | Death at 742$^{nd}$ day post-splenectomy |

ND–Not done

## Histological analysis

The findings from spleen histological analysis are summarized in Table 2. Patient VL3 exhibited a well-organized spleen (type 1) with a hyperplastic WP. Patient VL2 displayed a type 1 spleen with slightly organized WP and mildly atrophic lymphoid follicles. Patient VL1, however, presented with a type 3 spleen—disorganized and with atrophic lymphoid WP, featuring small follicles and a marginal zone (Fig 1). Remarkably, in this patient, the WP comprised only 5% of the total splenic tissue area. Plasmacytosis was either moderate or intense in the red pulp (RP) of all patients, slight in the periarteriolar lymphoid sheath (PALS) of patient VL3, and moderate in patients VL1 and VL2. Notably, infected macrophages were observed in all spleen compartments, consistently displaying high amastigote density.

## Leukocyte phenotype and cytokine-producing cells

All cell quantification data following immunohistochemistry labeling are presented in Table 3. Notably, a noteworthy observation is the decreased density observed across some leukocytes population, including cytokines, FoxP3, and caspase 3-expressing cells, both in the WP and the RP of patient VL1, whose spleen was disorganized. Surprisingly, only the percentage of CD3, CD4, CD8, CD68, and counts of IL6-positive cells were either similar or higher compared to those observed in patients with organized spleens.

Furthermore, it's worth noting that the density of B cells, IL-17, and TNF-producing cells was higher, while CD68+ macrophages appear to be lower in the patient without HIV infection compared to those in patients with HIV infection.

**Table 2. Histological analysis of spleen from VL patients.**

| PARAMETER | PATIENTS | | |
|---|---|---|---|
| | VL1 | VL2 | VL3 |
| Spleen size below left costal edge (cm) | 18 | 12 | 4 |
| Spleen weight (g) | 1882 | 1066 | 742 |
| White pulp: Size (%) | 5.0 | 10.0 | 21.7 |
| Organization (type) | 3 | 1 | 1 |
| Lymphoid follicle[1]: Size | -2.5 | -0.7 | 2.0 |
| Germinal center size | -2.0 | -0.7 | 3.0 |
| Infected macrophage | 1.3 | 0.3 | 3.0 |
| Hyaline deposits | 1.0 | 1.0 | 0.0 |
| Marginal and peri-follicular zone[1]: Size | -2.3 | -0.3 | 0.0 |
| Infected macrophages | 1.5 | 0.0 | 2.0 |
| Granuloma | 1.0 | 0.0 | 2.0 |
| Plasma cells | 1.0 | 0.0 | 0.0 |
| PALS[1]: Size | -0.5 | 0.3 | 0.0 |
| Infected macrophages | 1.8 | 2.0 | 1.3 |
| Plasma cells | 2.3 | 2.3 | 1.0 |
| Red pulp[1]: Cell density | -0.8 | 0.0 | 0.0 |
| Lymphocytes | 1.3 | 1.0 | 1.0 |
| Macrophages | 3.0 | 3.0 | 3.0 |
| Infected macrophages | 3.0 | 2.7 | 3.0 |
| Infection density | 3.0 | 3.0 | 3.0 |
| Granuloma | 0.5 | 0.0 | 1.3 |
| Plasma cells | 2.0 | 2.0 | 3.0 |
| Russel bodies/Mott cells | 1.3 | 1.7 | 1.0 |
| Neutrophils | 0.5 | 1.0 | 0.0 |
| Mast cells | 0.0 | 0.0 | 0.3 |
| Extramedullary hematopoiesis: Erythroblasts | 0.8 | 0.0 | 0.3 |
| Megakaryocytes | 0.8 | 0.7 | 0.0 |
| Sinus size | 2.3 | 2.0 | 1.0 |
| Iron deposits | 0.3 | 0.3 | 0.0 |

[1] semiquantitative scores

## Gene expression in VL patients

In this study, spleens from three patients with severe visceral leishmaniasis underwent evaluation for the expression of 590 genes using the nCounter platform. We incorporated the spleen transcriptome data from three normal donors obtained from publicly available sources (GTEx project) to serve as a control group. Our findings reveal distinct expression patterns of these genes between VL patients (n = 3) and the control group (n = 3), as depicted in the heatmap shown in Fig 2A. Additionally, through an unsupervised analysis using principal component analysis (PCA), we observed a notable disparity in gene signatures between patients with VL and spleen type 1 (VL3 and VL2) compared to the patient with spleen type 3 (VL1), as illustrated in Fig 2B.

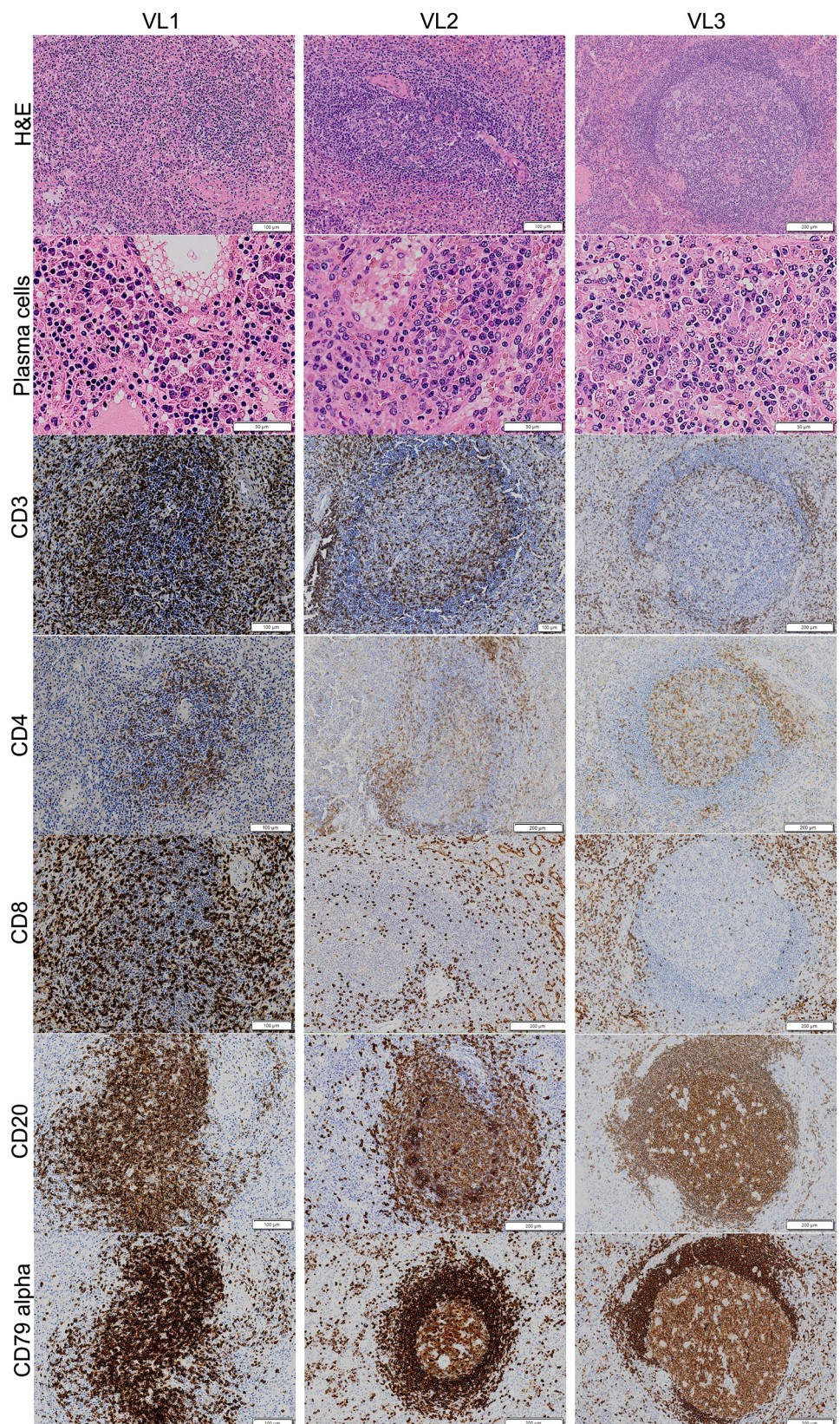

**Fig 1. Histological and leukocyte distribution in spleen from patients with VL.** Patient VL1 presents a spleen type 3, with atrophy of lymphoid follicles. Patient VL2 presents a spleen type 1, with slight organized lymphoid follicle. Patient VL3 presents a well-structured WP and well-defined sub compartments, classified as spleen type 1.

We identified 168 differentially expressed genes (DEGs) and most of these genes play pivotal roles in B cell proliferation and germinal center formation. Upregulated genes like *BCL10* and *ICOSLG* were notably prominent in VL2 and VL3 patients, while genes such as *CCR7* and *RIPK1*, associated respectively with T cell migration and apoptosis induction, exhibited downregulation, particularly in VL1 and VL3 patients (S3 Table).

**Table 3. Cell distribution in spleen compartments.**

| Parameters | VL1 | VL2 | VL3 |
|---|---|---|---|
| **White Pulp** | | | |
| CD3 (%) | 31 | 27 | 11 |
| CD4 (%) | 16 | 20 | 7 |
| CD8 (%) | 33 | 10 | 11 |
| CD20 (%) | 28 | 60 | 34 |
| CD79α (%) | 30 | 57 | 43 |
| CD68 (cell/mm$^2$) | 547 | 151 | 173 |
| IL1β (cell/mm$^2$) | 15 | 10 | 0 |
| IL4 (cell/mm$^2$) | 0 | 4 | 8 |
| IL6 (cell/mm$^2$) | 37 | 10 | 15 |
| IL10 (cell/mm$^2$) | 0 | 4 | 3 |
| IL17 (cell/mm$^2$) | 44 | 85 | 60 |
| TNFα (cell/mm$^2$) | 18 | 92 | 47 |
| TGFβ (cell/mm$^2$) | 0 | 19 | 0 |
| IFNγ (cell/mm$^2$) | 0 | 3 | 13 |
| Caspase 3 (cell/mm$^2$) | 8 | 19 | 140 |
| Foxp3 (cell/mm$^2$) | 19 | 136 | 198 |
| **Red Pulp** | | | |
| CD3 (%) | 21 | 12 | 13 |
| CD4 (%) | 4 | 2 | 4 |
| CD8 (cell/mm$^2$) | 2264 | 547 | 1586 |
| CD20 (cell/mm$^2$) | 593 | 683 | 896 |
| CD68 (cell/mm$^2$) | 1154 | 861 | 898 |
| CD79α (cell/mm$^2$) | 767 | 1896 | 1682 |
| IL1β (cell/mm$^2$) | 1 | 6 | 1 |
| IL4 (cell/mm$^2$) | 0 | 0 | 8 |
| IL6 (cell/mm$^2$) | 47 | 37 | 42 |
| IL10 (cell/mm$^2$) | 0 | 0 | 10 |
| IL17 (cell/mm$^2$) | 23 | 55 | 294 |
| TGFβ (cell/mm$^2$) | 0 | 5 | 0 |
| IFNγ (cell/mm$^2$) | 0 | 2 | 34 |
| TNFα (cell/mm$^2$) | 121 | 324 | 101 |
| Caspase 3 (cell/mm$^2$) | 8 | 17 | 18 |
| Foxp3 (cell/mm$^2$) | 1 | 885 | 511 |

Values of leukocytes and cytokines stained by immunohistochemistry. For some leukocytes were used percentual of area stained per white pulp or red pulp. These values represent the mean of 5 different areas per case.

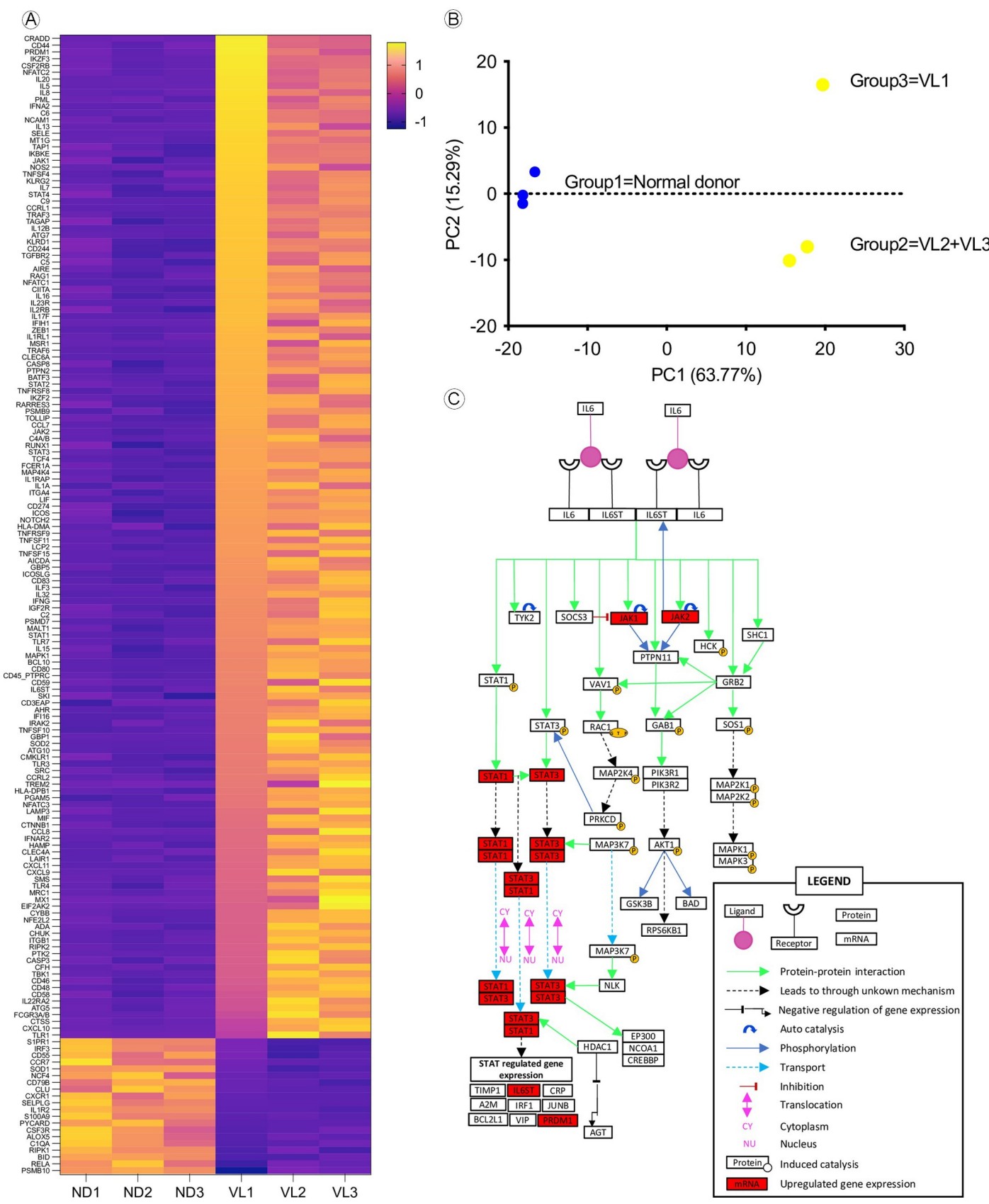

**Fig 2. Gene expression analysis of spleen from patients with VL.** A) Heatmap demonstrate that gene signature differs from patients with VL to normal donors and each band correspond to a different gene. Scale bar represents Log2 of Fold Change. Data were organized based on the gene expression of VL1 from the most expressed to the last expressed. B) Using an unsupervised analysis, we demonstrate that gene profile differs among VL patients, clustering organized spleen (patients VL2 and VL3) separated from disorganized spleen (patient VL1). Each dot represents the PCA value of each patient based on PCA analysis. Blue dot–Normal donor; Yellow dot–VL patient. C) IL6 signaling pathway adapted from Kandasamy and colleagues (2010) [18]. This pathway represents the signaling mechanism of IL6 in humans, genes upregulated in the spleen of patients with VL found on our analysis are highlighted in red, other genes presented at this graphic did not present expression change statistically significant or were not analyzed in our data set.

Our analysis of Canonical Pathways revealed an upregulation of genes involved in signaling through IL6 in VL patients, particularly evident in VL1 patient, including genes such as *JAK1*, *JAK2*, and *PRDM1* (Fig 2C).

## Integrative analysis

Aiming to identify differences between parameters in patient with VL only (patient VL2) or VL with HIV coinfected (patients VL1 and VL3), we performed a supervised analysis. Data were normalized and clusterized by Pearson correlation. The of number of *counts* per genes, laboratory parameters, and cell counting in IHC presented in S2 Fig.

This analysis revealed group of parameters more associated to each patient: To the patient VL2 without HIV coinfection and organized spleen the cluster with genes as CXCL9, *IL15*, *SOD2*, *IL32*, *GBP1*, *IRAK2*, *RIPK1*, *S100A9*, *IL6ST*, *CD45_PTPRC*, the cells CD20+ and TGFβ + in WP, cytokines IFNγ, TNFα, and IL1β in RP.

Among patients coinfected with both VL and HIV, we identified two distinct spleen types. Notably, the organized spleen type and VL-HIV coinfected was associated to genes as *AHR*, *BID*, *C1QA*, *CCL8*, *CCRL2*, *CLEC4A*, *CMKLR1*, *CSF3R*, *EIF2AK2*, *ICOSLG*, *ILF3*, *LAMP3*, *MRC1*, *PSMD7*, *SMS* and *SRC;* cells CASP3+, IL4+ and IFNγ+in the WP, and the cells IL4+, IL10+, IL17+, TGFβ+, CD20+ in the RP. Distinctly, VL1 patient, that has disorganized spleen and VL-HIV coinfection was related to *SOD1*, *STAT3*, *STAT4*, *ICOS*, *TRAF6* and *IRF3* genes, and CD8+ cells in the RP and expression of IL6 in both regions of spleen.

## Discussion

In this study we took the advantage of accessing an unusual specimen of spleens removed from patients with a severe form of VL, unresponsive to conventional therapeutics. The availability of these specimens allowed to confirm that human VL evolves with stages of splenic lymphoid hyperplasia and atrophy and may evolve to WP disruption and changes in leukocyte populations in the spleen compartments. These changes have already been observed in dogs naturally infected with *L. infantum* and in hamsters with experimental VL [15,19]. Although this small case series is not suitable for definitive conclusions, it highlights aspects that are supported by previous experimental observations, deserving to be explored in future studies using a larger series. For instance, as shown in the study by Silva and colleagues (2012) [20], in dogs with severe VL atrophic changes and WP disruption were intense in the lymphoid follicles and marginal zones and was almost unperceived in PALS. Similar predominance of changes in the lymphoid follicle was observed in the patient VL1 (with type 3 spleen).

Also, in agreement with the studies performed in dogs, herein we show a decrease in the number of B lymphocytes in the WP compartments and an increase of plasma cells in the RP. The reason of why plasma cell persists even after WP disruption is still not completely elucidated. In dogs with VL and disorganized spleen there is up-regulation of BAFF, APRIL and CXCL12, factors associated with extended survival and homing of plasma cells [21,22]. However, in this study we were not able to find differences in BAFF and APRIL expression, between the normal reference spleens and the spleens of infected patients. Furthermore, some

of these plasma cells may accumulate in the RP following an extrafollicular differentiation pathway [23].

The gene expression pattern seen in the spleen of patients with VL was different from that observed in the normal spleens used as reference. Despite of analysis of bulk RNA spleen samples did not allow to understand how genes are being expressed at cellular level, in this work we observed that the pattern of DEGs in VL are related to inflammatory response, cellular and blood cell homeostasis and signaling by the B cell activator factor and IL6 cytokine. These pathways inflammation process and cell cycle process are highly expressed in blood and spleen of hamsters with experimental visceral leishmaniasis [24,25].

Among the investigated genes and cellular characterization by IHC, we found that *IFNG* gene appeared highly expressed in VL patients, as observed in other transcriptional studies in human and experimental VL [24–26], and the gene expression pattern in VL3 patient parallels to the number of IFNγ + cells present in the spleen. However, only a few cells stained for IFNγ + were observed in the IHC staining of the spleen of the patients in this and in a previous study [27]. It is still not clear the difference between gene expression and IHC finding. The fact that all these patients presented treatment failure and relapses of disease, they also presented infected macrophage intensely distributed through the spleen compartments. The failure in parasite growth control may result of a local failure in the pathways of IFNγ expression, especially in disorganized spleen. In fact, cells producing the regulatory cytokines IL4, IFNγ, IL10 and IL17 were detected or more frequently found in the patient with VL and hyperplastic spleen.

In this series, the spleen of the patient VL1 (spleen type 3, disorganized), showed an increased density of CD8+ lymphocyte in the WP not observed in the other patients. This change was not also observed in the studies with dogs [20,28]. Further studies are, therefore, necessary to confirm the relevance of this finding in spleen disorganization in VL.

There was an overall increase of CD68+ macrophage in VL1 patient compared with the patients with VL and organized spleen. Most of the cytokine-producing cell distribution and caspase 3 or Foxp3-expressing cells revealed a trend towards a decrease in density in the WP or RP in the patient with VL and histologic disorganization of the spleen. This finding together with the absence of cells that stained for IL10, IL4, IFNγ, or TGFβ may reflect a state of T cell exhaustion as it has been reported in the spleen of dogs with VL [28].

The spleen of patient VL1 had a unique pattern of gene expression different from that observed in the spleen type 1 of the other two patients. It showed high expression of genes from IL6 signaling pathway (*JAK1*, *JAK2* and *PRDM1*), and a high density of IL6-producing cells in the RP and the WP, differing from organized spleen, suggesting that this pathway can also be involved in spleen disorganization. IL6 is involved in the induction of inflammatory processes and is found at high levels in plasma of humans and dogs infected with *Leishmania infantum* [29–31]. The IL6 levels have been proposed as a biomarker of prognosis and death in VL [13,32]. One of the outcomes of IL6 signaling is the induction of *PRDM1* expression. This gene was found to induce B cell differentiation into plasma cell, to be highly expressed in T cell exhaustion and to induce regulatory FOXP3+ T cell to secrete IL10 [33,34]. T cell exhaustion is a condition that has been associated with the progression of VL and spleen disorganization [28,35,36], particularly in patients with HIV co-infection [4]. The patients with VL of this study showed high expression levels of *PRDM1*, *CD274* (*PD-L1*), and *CD44*, which also supports the hypothesis of T cell exhaustion. Conversely, some DEG's, such as *BCL10* and *ICOSLG*, were highly expressed in type 1 spleen. These genes are associated with co-stimulatory T-cell and B cells proliferation and are involved in B cell differentiation to plasma cells. It is interesting to notice that plasmacytosis establishes since the hyperplastic stage of spleen in VL and remains until the late stage even when the WP is already disrupted [23]. Therefore,

these gene expression patterns may correspond to the lymphoid follicle hyperplasia and spleen plasmacytosis observed in some stages of VL. This process of differentiation could be due to TNFα, a cytokine that induce *ICOSLG* activation and is increased in the type 1—hyperplasic spleen as seen in our patients and in experimental studies [37,38].

Despite patients coinfected VL-HIV having indetectable viral copies in the spleen, changes due to infection may persist and have impact on the outcome of the disease. In fact, there is a percentage of people living with HIV that become immunological non-responders, presenting low CD4 counts even after reduction of viral load [39]. The patients with VL-HIV presented low density splenic CD4$^+$ cells in the WP, low expression of genes such as *IL15* and of cytokines related to Th1 response in comparison to the patient with VL monoinfection, also, literature has associated upregulation of genes related to Th2 response and downregulation of *IFNG* gene before VL treatment in patients VL-HIV coinfected [40]. This observation concurs with previous findings, in which patients with VL-HIV presented more frequent disease relapse, associated to low CD4 counts and low IFNγ in circulating blood [41]. Conversely, IL15 a cytokine associated with induction of Th1 response through IFNγ signaling, contribute to cure in canine VL [42], is upregulated in the patient with VL only. These differences in gene expression patterns and cellular distribution may be distinctive of coinfection status.

The patients with HIV infection (VL1 and VL2) presented extreme changes of the spleen (atrophy and disorganization in patient VL1 and hyperplastic changes inpatient VL3). This may be indicative of dysregulated activation of immune system in LV-HIV infection that may eventually lead to lymphoid tissue disruption and immune system exhaustion. Both conditions VL and HIV infection are both conditions known to be related with WP disruption. On the clinical point of view, these patients also had little improvement of hematological parameters after splenectomy and evolved to death.

The main limitation on this study is the number of subjects. However, to our knowledge, this is the first study combining morphology (conventional and IHC) and transcriptomic analysis and IHC to characterize human spleen changes present in visceral leishmaniasis. The data presented here support the following hypotheses: 1) VL also cause disorganization of spleen compartments in humans. 2) WP disruption is associated to the phenomenon of immune system exhaustion; 3) IL6 may play a role in disorganization and plasma cell accumulation in the spleen.4) low CD4 T cell density and low IL15 expression may contribute to ineffective response to the parasite favoring recurrence of the disease and eventually WP disruption; 5) impaired IFNγ signaling may lead to persistence of *Leishmania* and chronic inflammation, thereby altering the spleen's microenvironments and disrupting immune response to infections.

## Material and methods

### Ethical statement

Written informed consent was obtained from the patients, for publication of this case series as stated in a previously published paper [17].

### Clinical and laboratory data

Information was gathered from three adult patients experiencing relapsing VL, showing resistance to standard Sb$^V$ or amphotericin B treatments, and subsequently undergoing therapeutic splenectomy. Their clinical records were meticulously examined, focusing on key variables: age, gender, disease duration, HIV co-infection, spleen dimensions, fever occurrence, red and white blood cell counts, platelet levels, serum protein levels, as well as any co-infections pre- and post-splenectomy.

## Spleen samples

Upon splenectomy, tissue fragments were promptly retrieved from the midsection of the spleen by making a transverse incision across the capsule and the larger axis of the spleen. These fragments were then prepared for histological analysis and gene expression studies in the following manner:

## Spleen conventional histology

The collected spleen fragments underwent fixation in formalin-acetic acid-alcohol solution, followed by embedding in paraffin. Tissue sections, 4 micrometers thick, were stained using hematoxylin and eosin, as well as periodic acid–Schiff. Subsequently, two pathologists (WLCS and LARF) conducted examinations, classifying the samples into three categories (spleen type 1, type 2, and type 3) based on the structural organization of the splenic WP. These categories were determined according to established criteria [9,19]: spleen type 1 exhibiting well-organized, reactive WP with distinct peri-arteriolar lymphocyte sheaths and multiple reactive lymphoid follicles characterized by well-defined germinal centers, mantle zones, and marginal zones.; spleen type 2 containing primarily inactive lymphoid follicles, displaying slightly disorganized WP with hyperplastic or hypoplastic changes, resulting in less discernible boundaries between WP compartments.; spleen type 3 Featuring extensively disorganized WP, often presenting poorly discernible compartments, frequent lymphoid atrophy, and a lack of reactive lymphoid follicles.

Cells were morphologically characterized according to previously described criteria: Lymphocytes–cells with round nuclei containing condensed chromatin, inconspicuous nucleoli, and scant cytoplasm. Plasma cells–cells with eccentric nuclei, heterochromatin dispersed around the edge in a pattern like that of the numerals on an analog clock face, and basophilic cytoplasm with clear perinuclear vacuoles. Macrophages–large cells with oval or reniform nuclei containing loosely packed chromatin surrounded by a rim of eosinophilic cytoplasm with indistinct edges. Granulocytes–cells with highly lobulated nuclei containing densely packed chromatin. Megakaryocytes–very large cells with wide and clearly defined cytoplasm and large, lobulated nuclei containing condensed chromatin.

Furthermore, immunohistochemistry was employed for further characterization. Initially, the density of different cell populations within the RP was scored as follows: 0 = no cells, 1 = isolated cells or small aggregates in a small proportion of the examined ×400-magnified microscopic fields, 2 = single cells or aggregates observed in most examined ×400-magnified microscopic fields, and 3 = single cells or aggregates observed in nearly all examined ×400-magnified microscopic fields. These observations were validated through morphometric analysis.

**Leukocyte populations and cytokine producing cells.** Immunophenotyping and identification of cytokine-producing cells was performed by immunohistochemistry using the following antibodies and procedures: Anti-human molecules antibodies: CD3 (polyclonal produced in rabbit), CD4 (4B12), CD8 (C8/144B), CD20 (L26), CD68 (KP1), CD79α (JCB117), all from Dako (Glostrup, Denmark). Procedures: Deparaffinization and rehydration were performed in xylene, followed by a decreasing series of ethanol solutions and the blockage of endogenous tissue peroxidase in 3% hydrogen peroxide and antigen retrieval was performed using a Dako PT Link (PT100/PT101) high pH module at 97˚C for 20 minutes. The DAKO EnVision + HRP kit (Dako Co, Denmark) was used as an amplification system. Diaminobenzidine was used as the chromogen. Sections were counterstained with hematoxylin. Mouse IgG1 isotypes unrelated to the antigens were used as a negative control to replace the primary antibody, except for CD3, that negative control was only incubated with secondary antibody only.

Cytokine detection was performed using these primary antibodies: TGFβ (ab66043), IL1β (AF-201-NA), IFNγ (MAB 285), TNFα (AF-210-NA), IL10 (AF-217-NA), IL6 (AF-206-NA), IL17 (AF-317-NA) and IL4 (AF-204-NA) (RD Systems), as well as 14-4776-82 (Foxp3 from E-Bioscience) and Caspase 3, in accordance with the following protocol: Antigen unmasking was performed using a heat-induced antigen retrieval method in a water bath with Retrieval Buffer (Dako Corporation, Carpinteria, CA, USA) for 25 minutes at 95˚C, pH 9.0. Next, the sections were incubated in a saponin solution (0.1% in PBS 0.01 M, pH7.4) for 10 minutes at room temperature, followed by incubation in skim milk 10% for 30 minutes and a final incubation with the primary antibodies diluted in 1% bovine albumin–PBS solution, overnight at 4˚C. Specific antibody binding was detected using a second antibody and the LSAB system (Dako Corporation, Carpinteria CA, USA, K690) for 30 minutes at 37˚C. All reactions were developed using a 3′3 diaminobenzidine chromogen solution and counterstained with Harris hematoxylin. The density of leukocyte cell populations was estimated by morphometry. Since cytokine-producing cells were distributed in small clusters the density of cytokine producing cells was estimated by identifying the two most dense sites of cell clustering and count the cells in this site and in other four non-overlapping areas around this site.

## Human spleen transcriptomics

**Sample from VL spleen tissue.**   Total mRNA was initially extracted from tissue fragments using a modified two-step protocol that involved extraction with TRIzol Reagent, followed by further purification using Spin Column-Based technology. Specifically, we macerated tissue fragment samples with 800 μL of TRIzol Reagent (Thermo Fisher Scientific, Cat. no. 15596026) followed by the addition of 160 μL of chloroform (Sigma-Aldrich, Cat. no. 516726), then centrifuged the mixture for 15 minutes at 12,000 x g at 4˚C. The obtained aqueous phase was then subjected to additional purification using Spin Column-Based technology (RNeasy Mini Kit 50, Cat. No. / ID: 74104), to ensure the extraction of high-quality total mRNA. The quantity of RNA was measured using NanoDrop (ThermoFischer). The quality of RNA was assessed using Agilent Bioanalyzer automated electrophoresis (Agilent) and considered viable if the RNA integrity number (RIN) was greater than 7. Gene expression analysis was conducted using the nCounter platform by NanoString Technologies, which employs molecular barcodes directly linked to target RNA transcripts, allowing for digital detection. This analysis was performed at the Genomics Core Leuven (VIB/KULeuven—Belgium).

For this assay, two probes were utilized: one to capture specific mRNA through complementary binding and another linked to a fluorescent barcode, aiding in the identification of the target mRNA through hybridization. The combination of these probes with genetic material enables the detection of desired transcripts via fluorescent barcodes.

**Sample from healthy spleen tissue.**   As a control, healthy spleen tissue samples were reanalyzed from the GTEx project [43], which conducted whole-transcriptome profiling of various human tissues, including spleen. Three samples were chosen (S1 Table) based on age and gender similarity to the profile of infected patients. The biomaRt library for R [44] was utilized for gene annotations. Raw reads were mapped against the human genome reference (GRCh38) using STAR [45] with default parameters. HT-seq count [46] assigned and counted mapped reads to annotated genomic features based on GENCODE v. 25 annotations.

**Differentially Expressed Genes (DEGs).**   Normalization across samples (S1 Fig) was performed multi-step: First, the expression values of all housekeeping genes (S2 Table), $\hat{s}$, were averaged across samples by using the geometric mean. Then, lane scaling factors $\hat{c}_i$ were obtained by estimating a combined term as follows: $\hat{c}_i = \frac{\sum_{i=1}^{n} \hat{s}_i}{n}$, where i represents each sample. A normalized expression value $\hat{g}$ was estimated for every gene in each sample $g$ by

calculating $\hat{g} = g * \hat{c}_i$. The DEGs data analysis employed an unpaired t-test, assuming individual row variance, and adjusted for False Discovery Rate (FDR) at 1.00% using the two-stage step-up method by Benjamini, Krieger, and Yekutieli.

A heat map (GraphPad Prism Software) was used to display the significant Differentially Expressed Genes (DEGs), illustrating their distribution by converting gene counts into Z-scores. Principal Component Analysis (PCA) was used to reduce the dimensionality of the measured genes in the dataset, effectively capturing the disease conditions of the experiment (Weka 3.9.3, University of Waikato).

## Supporting information

**S1 Table. GTEx samples used as healthy liver tissue control.**
(PDF)

**S2 Table. Housekeeping genes used during normalization.**
(PDF)

**S3 Table. Differentially Expressed Genes in Normal and VL infected patients.** Each value represents a Log2FC for each gene. ND–Normal Donor; VL–visceral leishmaniasis infected patient.
(XLSX)

**S1 Fig. Genes expression level distribution after normalization.**
(PDF)

**S2 Fig. Heatmap of data from VL patients.** Each row represents each patient (VL1, VL2, VL3), while the number of counts per gene (G), number of cells stained by IHC (C) in WP or RP and laboratorial data (L) are present in columns. The columns were clustered based on Pearson correlation data to each patient. WP- White pulp; RP- Red pulp.
(PDF)

## Author Contributions

**Conceptualization:** Ricardo Khouri, Carlos Henrique N. Costa, Washington L. C. dos-Santos.

**Data curation:** Luis Gustavo C. Reinaldo, Pablo Ivan P. Ramos, Washington L. C. dos-Santos.

**Formal analysis:** Jonathan L. M. Fontes, Ricardo Khouri, Caroline V. B. de Melo, Pablo Ivan P. Ramos, Johan Van Weyenbergh, Luiz A. R. de Freitas, Washington L. C. dos-Santos.

**Funding acquisition:** Washington L. C. dos-Santos.

**Investigation:** Jonathan L. M. Fontes, Ricardo Khouri, Carla Pagliari, Marta Santos, Raimundo José C. Araújo, Jr., Johan Van Weyenbergh, Luiz A. R. de Freitas.

**Methodology:** Jonathan L. M. Fontes, Luis Gustavo C. Reinaldo, Erina M. A. Hassegawa, Antônio José Meneses Filho, Caroline V. B. de Melo, Rafael de Deus Moura, Carla Pagliari, Johan Van Weyenbergh, Luiz A. R. de Freitas, Carlos Henrique N. Costa.

**Project administration:** Washington L. C. dos-Santos.

**Resources:** Jonathan L. M. Fontes, Antônio José Meneses Filho, Rafael de Deus Moura, Carla Pagliari, Raimundo José C. Araújo, Jr., Washington L. C. dos-Santos.

**Supervision:** Washington L. C. dos-Santos.

**Visualization:** Jonathan L. M. Fontes.

**Writing – original draft:** Jonathan L. M. Fontes, Ricardo Khouri, Washington L. C. dos-Santos.

**Writing – review & editing:** Jonathan L. M. Fontes, Carlos Henrique N. Costa, Washington L. C. dos-Santos.

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
