## [Decision Letter · Decision Letter 0]

1 Feb 2024

Dear Mr. Fontes,

Thank you very much for submitting your manuscript "Tittle: An integrated analysis of the structural changes and gene expression of spleen in human visceral leishmaniasis with and without HIV coinfection" for consideration at PLOS Neglected Tropical Diseases. As with all papers reviewed by the journal, your manuscript was reviewed by members of the editorial board and by several independent reviewers. In light of the reviews (below this email), we would like to invite the resubmission of a significantly-revised version that takes into account the reviewers' comments. 

The study is an important advancement in understanding the dynamics of HIV/VL co infection in real patient samples. Please respond to reviewers comments and concerns especially the interpretation of the data as well as the parameters studied.

We cannot make any decision about publication until we have seen the revised manuscript and your response to the reviewers' comments. Your revised manuscript is also likely to be sent to reviewers for further evaluation.

Sincerely,

Hira L Nakhasi, Ph.D.

Section Editor

Abhay Satoskar

Section Editor

The study is an important advancement in understanding the dynamics of HIV/VL co infection in real patient samples. Please respond to reviewers comments and concerns especially the interpretation of the data as well as the parameters studied.

Reviewer's Responses to Questions

**Key Review Criteria Required for Acceptance?**

**Methods**

-Are the objectives of the study clearly articulated with a clear testable hypothesis stated?

-Is the study design appropriate to address the stated objectives?

-Is the population clearly described and appropriate for the hypothesis being tested?

-Is the sample size sufficient to ensure adequate power to address the hypothesis being tested?

-Were correct statistical analysis used to support conclusions?

-Are there concerns about ethical or regulatory requirements being met?

Reviewer #1: (No Response)

Reviewer #2: Methods were well described.

Reviewer #3: The objectives are stated clearly, however, the data are limited and need to be described better to confirm the hypothesis.

The small sample size (3 patients) is adequate for a descriptive clinical study.

Potentially proper statistics are used, they need better description 

No concerns about ethics or regulations.

**Results**

-Does the analysis presented match the analysis plan?

-Are the results clearly and completely presented?

-Are the figures (Tables, Images) of sufficient quality for clarity?

Reviewer #1: (No Response)

Reviewer #2: Results are well described.

Reviewer #3: The analysis matches the plan.

The results are extensively presented. However, they are not complete. Gene expression was described as transcripts per gene, yet those numbers were not presented.

The tables are all acceptable. The figures require further explanation.

**Conclusions**

-Are the conclusions supported by the data presented?

-Are the limitations of analysis clearly described?

-Do the authors discuss how these data can be helpful to advance our understanding of the topic under study?

-Is public health relevance addressed?

Reviewer #1: (No Response)

Reviewer #2: (No Response)

Reviewer #3: The conclusions are largely descriptive consistent with the authors' awareness that the study has few samples. The conclusions drawn often refer to select data to make statements that are not supported by all the data.

The authors focus on stating that the findings in human patients confirm previous results in animals.

The public health benefit is discussed in terms of contribution to the understanding of Leishmania pathogenesis.

**Editorial and Data Presentation Modifications?**

Reviewer #1: (No Response)

Reviewer #2: (No Response)

Reviewer #3: See attached review memo.

**Summary and General Comments**

Reviewer #1: (No Response)

Reviewer #2: Strength # 

1. This study was performed with rare clinical tissue samples, particularly HIV along with Visceral leishmaniasis.

2. Authors used very well acceptable approach to tease out gene expression at mRNA level.

Weakness # 1. As author mentioned in the discussion section about the low number of samples. Nevertheless, still this study will be a good addition for the Leishmania field as there is not much study report regarding HIV-Leishmania co-infection in endemic region.

Overall, study design to analyze the tissue sample and result analysis was well performed. Please see below my comments:

1. Authors have performed cellular distribution analysis with immuno-histochemical analysis. However, as this technique is very challenging and is varies to one antibody marker to another antibody, the Results depicted in Table 3 needs further validation. Cytokine expression level measured from the serum samples of these patient would be an ideal confirmation of these results.

2. IL-6 signaling is driving force for the IL-17 signaling pathway, which further modulates several inflammatory pathways involve in Visceral Leishmaniasis. Please provide the outcome of gene expression analysis of IL-17 pathway.

3. Please provide an access to the Nano string data analysis for the readers or submit the gene expression analysis in public database upon publication of this manuscript.

Reviewer #3: See attached review memo.

PLOS authors have the option to publish the peer review history of their article (what does this mean?). If published, this will include your full peer review and any attached files.

Reviewer #1: Yes: Parna Bhattacharya

Reviewer #2: Yes: Ranadhir Dey

Reviewer #3: No
---

## [Editor Report · Decision Letter 1]

30 Apr 2024

Dear Dr. dos-Santos,

We are pleased to inform you that your manuscript 'Tittle: An integrated analysis of the structural changes and gene expression of spleen in human visceral leishmaniasis with and without HIV coinfection' has been provisionally accepted for publication in PLOS Neglected Tropical Diseases.

Best regards,

Abhay R Satoskar

Section Editor

Abhay Satoskar

Section Editor

---

## [Editor Report · Acceptance letter]

9 May 2024

Dear Dr. dos-Santos,

We are delighted to inform you that your manuscript, "Tittle: An integrated analysis of the structural changes and gene expression of spleen in human visceral leishmaniasis with and without HIV coinfection," has been formally accepted for publication in PLOS Neglected Tropical Diseases.

Best regards,

Shaden Kamhawi

co-Editor-in-Chief

Paul Brindley

co-Editor-in-Chief
